# Genetic characterization of primary and metastatic high-grade serous ovarian cancer tumors reveals distinct features associated with survival

Emilee N. Kotnik [1,2,3], Mary M. Mullen[1,2,3], Nicholas C. Spies[3,4], Tiandao Li [5], Matthew Inkman [6], Jin Zhang [6], Fernanda Martins-Rodrigues [7], Ian S. Hagemann [1,2,3,4], Carolyn K. McCourt[1,2,3], Premal H. Thaker[1,2,3], Andrea R. Hagemann[1,2,3], Matthew A. Powell[1,2,3], David G. Mutch[1,2,3], Dineo Khabele[1,2,3], Gregory D. Longmore [7,8], Elaine R. Mardis [9], Christopher A. Maher [7,10,11,12], Christopher A. Miller[7] & Katherine C. Fuh [1,2,3,13 ✉]

High-grade serous ovarian cancer (HGSC) is the most lethal histotype of ovarian cancer and the majority of cases present with metastasis and late-stage disease. Over the last few decades, the overall survival for patients has not significantly improved, and there are limited targeted treatment options. We aimed to better characterize the distinctions between primary and metastatic tumors based on short- or long-term survival. We characterized 39 matched primary and metastatic tumors by whole exome and RNA sequencing. Of these, 23 were short-term (ST) survivors (overall survival (OS) < 3.5 years) and 16 were long-term (LT) survivors (OS > 5 years). We compared somatic mutations, copy number alterations, mutational burden, differential gene expression, immune cell infiltration, and gene fusion predictions between the primary and metastatic tumors and between ST and LT survivor cohorts. There were few differences in RNA expression between paired primary and metastatic tumors, but significant differences between the transcriptomes of LT and ST survivors in both their primary and metastatic tumors. These findings will improve the understanding of the genetic variation in HGSC that exist between patients with different prognoses and better inform treatments by identifying new targets for drug development.

[1] Division of Gynecologic Oncology, Washington University in St. Louis, 660 S. Euclid Ave Mailstop, 8064 St. Louis, MO, USA. [2] Center for Reproductive Health Sciences, Washington University in St. Louis, 660 S. Euclid Ave Mailstop, 8064 St. Louis, MO, USA. [3] Department of Obstetrics and Gynecology, Washington University in St. Louis, 660 S. Euclid Ave Mailstop, 8064 St. Louis, MO, USA. [4] Department of Pathology and Immunology, Washington University in St. Louis, 660 S. Euclid Ave CB, 8118 St. Louis, MO, USA. [5] Department of Developmental Biology, Washington University in St. Louis, 660 S. Euclid Ave CB, 8103 St. Louis, MO, USA. [6] Department of Radiation Oncology, Washington University in St. Louis, 660 S. Euclid Ave CB, 8224 St. Louis, MO, USA. [7] Division of Oncology, Washington University in St. Louis, 660 S. Euclid Ave CB, 8069 St. Louis, MO, USA. [8] ICCE Institute, Washington University in St. Louis, 660 S. Euclid Ave CB, 8225 St. Louis, MO, USA. [9] Institute for Genomic Medicine, Nationwide Children's Hospital, 575 Childrens Crossroad, Columbus, OH, USA. [10] McDonnell Genome Institute, Washington University in St. Louis, 4444 Forest Park Avenue, CB 8501 St. Louis, MO, USA. [11] Department of Internal Medicine, Washington University in St. Louis, 660 S. Euclid Ave, MSC 8066-22-6602 St. Louis, MO, USA. [12] Department of Biomedical Engineering, Washington University in St. Louis, McKelvey School of Engineering, 1 Brookings Drive, St. Louis, MO, USA. [13] Department of Obstetrics and Gynecology & Reproductive Sciences, University of California San Francisco, San Francisco, CA, USA. ✉email: katherine.fuh@ucsf.edu

High-Grade Serous Cancer (HGSC) of ovary, fallopian tube, or peritoneum is the most lethal ovarian cancer histotype and the second most common gynecologic malignancy[1,2]. Over the past few decades, chemotherapy has been the standard of care, yet overall survival (OS) has not significantly improved[3,4]. PARP inhibitors, which target base-excision DNA repair mechanisms and cause genetic lethality in tumors of patients harboring BRCA1 or BRCA2 mutations or homologous recombination deficiencies, can be used as maintenance therapies, but are only applicable for ~ 50% of HGSC patients[5,6]. The majority of cases, about 80%, present with late-stage disease, when the tumor has already metastasized[3,4,7]. These patients have only a 29.2% chance of surviving longer than 5 years[4]. Therefore, to improve patient survival, we sought to better characterize the genomic and transcriptomic landscapes of matched primary and metastatic ovarian cancers and to identify novel targets for drug development, especially in a more aggressive metastatic disease setting.

Large-scale tumor characterizations, by consortia such as The Cancer Genome Atlas (TCGA), have established that primary HGSC tumors harbor ubiquitous TP53 mutations and copy number alterations, and a low prevalence of other recurrently mutated genes[8]. Prior genomic studies of ovarian cancers have only included large numbers of primary tumors rather than comparing matched primary and metastatic disease in the context of outcomes. A recent study, Yang et al., characterized clinical and genomic features from HGSC primary tumors that correlated with short-term (ST, OS < 2 years) and long-term (LT, OS > 10 years) survival[9]. While this study does provide evidence that there are clinical and genomic characteristics unique to patient survival in primary tumors, metastatic tumors were not included. Study design is further nuanced by the context of survival duration, since exceptional survivors (>10 years of survival), have a high prevalence of BRCA mutations, and are known to respond well to standard therapy[10]. One study has examined genomic and transcriptomic sequencing from matched primary and metastatic tumors in the context of response to chemotherapy or surgical resection[11]. Another study identified a transcriptome signature that distinguished between primary and metastatic tumors but did not relate this to survival[12].

Here, we sought to determine whether there are unique features in the genomes and transcriptomes of metastatic tumors from short-term survivors when compared to their matched primary tumors and/or to primary/metastasis paired tumors from long-term survivors. Our cohort design examines these differences between tumors from patients within the median survival time for ovarian cancer[13]. In this context, we compared somatic variants, copy number alterations, mutational burden, differential expression, immune cell infiltrates, and gene fusion predictions between chemo naïve primary and metastatic tumors from 23 HGSC short-term (ST, OS < 3.5 years) survivors and 16 HGSC long-term (LT, OS > 5 years) survivors using whole-exome sequencing (WES) and RNA sequencing (RNA-seq).

## Results

### Characterization of genomic landscape

We compared somatic variants, copy number alterations, and mutational burden between the primary and metastatic tumors of the ST and LT survival groups. Our cohort of patient tumors exhibited characteristics typical of those seen in previously sequenced HGSC tumors, such as nearly ubiquitous TP53 mutations, high numbers of copy alterations, and a low number of recurrently mutated genes (Fig. 1a, b). As was found in the Yang et al. study, our cohort of LT survivors also exhibited a significantly greater mutational burden than the ST survivors (Mann–Whitney–Wilcoxon test statistic =

611, p-value = 5.1e-6)[9]. There was no statistical difference between the mutational burden of paired primary and metastatic tumors (MW stat = 611, p-value = 0.4299) (Fig. 2c).

Interestingly, in comparison to recurrently mutated genes identified in the TCGA cohort, we also observed that RB1 mutations were found only in primary and metastatic tumors of LT survivors. This is a finding consistent with studies analyzing exceptional HGSC survivors[8,14]. CDK12 was only mutated in the LT survivors. We confirmed previously published findings that tumors of LT survivors were more likely to have BRCA1 alterations and copy-altered segments when compared to ST survivors[9]. In total, we identified an average of 723 somatic mutations per LT survivor (10,124 SNVs total/14 patients) and an average of 591 somatic mutations per ST survivor (13,599 SNVs total/23 patients). Four patients exhibited somatic BRCA1 mutations with VAF > 30 (Supplementary Table 2). Six patients exhibited germline BRCA1 mutations and 1 LT survivor had a benign germline BRCA2 mutation (VAF > 30), all of which had mixed ClinVar-based clinical interpretations of pathogenic significance (Supplementary Tables 3 and 4)[15].

We applied Classification of Ovarian Cancer signatures to our dataset and observed no statistically significant differences based on survivorship or tumor type among the signatures. The most prevalent signatures among our cohort were Mesenchymal and Differentiated (Fig. 1b)[16]. To compare the types of variants found in each of the tumors, we calculated the percentage of total variants identified in each patient that can be contextualized by the human cancer mutational signatures[17,18]. In our cohort, the most common signatures were for 5- methylcytosine deamination, mismatch repair, and double-strand break repair, along with a large number of mutations contributing to the unknown signature. There were no statistically significant differences between the signature percentages when comparing matched primary and metastatic tumors or ST and LT survivors. However, there is a higher percentage contribution to the mismatch repair signature in LT survivors compared to ST survivors, although not statistically significant (Supplementary Fig. 4). We also calculated percentages of the mutational nucleotide transitions and transversions. Transitions from C > T account for the largest percentage of total mutations in most of the tumor samples. The percent of total mutation ratios remain relatively the same between the primary and metastatic tumors in the majority of patients, but large shifts in the mutational transitions and transversions can be seen in Patients 15, 20, 30 and 34. There were no statistically significant differences when comparing the mutational percentages of transitions and transversions between ST and LT survivors and primary and metastatic tumors (Supplementary Fig. 5).

### ST survivors exhibited a higher percentage of shared variants

For each patient, we calculated the percentage of called variants that were unique to the primary or to the metastatic tumor, or were shared between the two tumors. We observed that there were higher percentages of shared variants between the primary and metastatic tumors for ST survivors compared to LT survivors, although this was not statistically significant (MW statistic = 117.0, p-value = 0.0866) (Fig. 2a, b). Of note, all of our LT survivor samples were FFPE whereas the ST survivors included FF specimens. There was no FFPE/FF specific variant filtering applied in our variant calling pipeline, but each sample did undergo quality control and log-likelihood ratio (LLR) filtering (https://github.com/genome/docker-somatic-llr-filter/blob/master/somatic_llr_filter.py). Thus, the differences seen in the number of shared variants of the ST and LT cohorts could be affected by FFPE artifacts from sample preparation.

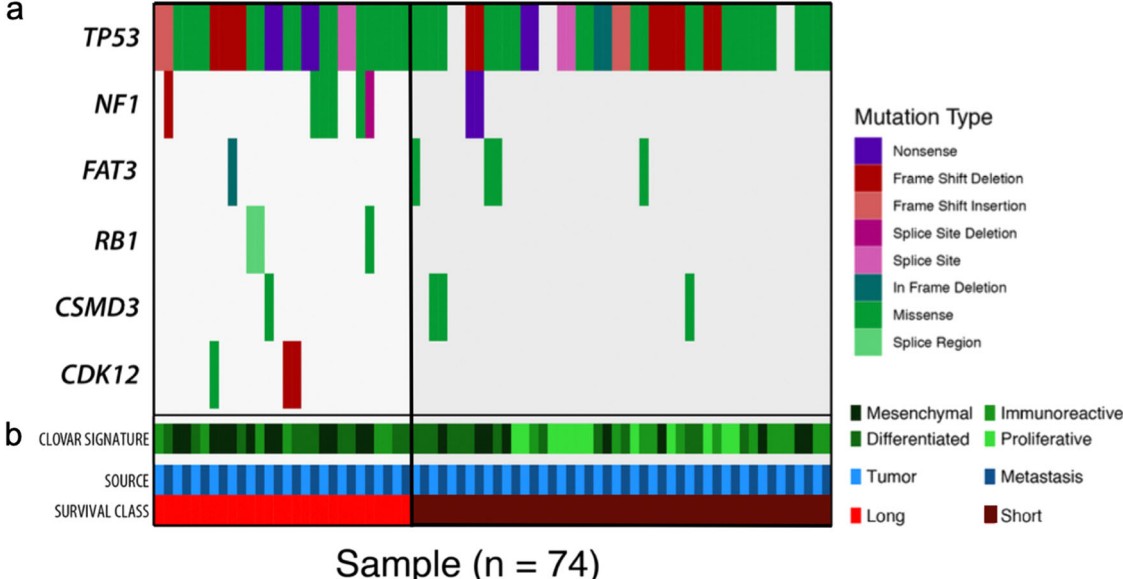

**Fig. 1 Mutational Landscape of primary and metastatic tumors from ST and LT survivors. a** Samples ($n = 74$) are organized as matched tumor pairs, with primary first followed by metastatic tumor. Somatic mutations in OV TCGA frequently mutated genes, colors indicate mutational type. **b** Status of samples source (primary tumor (light blue), metastatic tumor (dark blue)), survival class (ST (dark red), LT (red)), and Classification of Ovarian Cancer signature (mesenchymal, differentiated, immunoreactive, and proliferative) (Green shades).

Within our cohort, all but 2 patients had tumors that harbored *TP53* variants (Supplementary Table 5). Of the 35 patients that carried *TP53* mutations, all but one patient shared the same *TP53* mutation between their primary and metastatic tumor and 4 patients carried multiple *TP53* mutations. The majority of *TP53* mutations were missense or frame-shift deletions within the DNA-binding domain. One known hotspot mutation, R273H, was present in 3 patients, 2 of which were ST survivors. The 2 ST survivors that harbored this hotspot mutation had an overall survival ranging between 17-19.6 months, whereas the LT survivor lived more than 147 months after their diagnosis. In the TCGA-OV patient cohort, 2% (11/489) of tumor samples also had the *TP53* R273H mutation, compared to the 8% (3/37) in our cohort[8,19,20].

**LT survivors exhibited more copy number alterations**. Concordant with findings from the TCGA-OV project, CNAs were abundant in these data, with CNAs observed in every sample, and the number of copy-altered segments ranged from 33 to 739, with segment lengths ranging from 3636 to 229,754,969 nucleotides. The average segment length was 3,997,903 nucleotides (Supplementary Fig. 1a–c)[8]. The LT survivors had a greater proportion of copy-altered segments ($p = 0.03$, 95% CI 0.004 to 0.11), driven by a greater proportion of amplifications ($p = 0.01$, 95%CI 0.01–0.1). There was no significant difference between primary tumor samples and metastases, nor were there significant differences in mean estimated ploidy between groups.

We identified recurrent CNAs in our cohort overall and subsets of the ST survivors, LT survivors, primary tumors, and metastatic tumors (Supplementary Fig. 2a–c)[21]. Overall, our cohort exhibited a total of 254 recurrent copy-altered segments, including 85 amplified segments and 169 deleted segments with a 90% confidence interval. We identified 2333 genes within the amplified peaks and 4904 genes within the deleted segments. Region 20q13.12, previously identified in other ovarian tumors, was amplified in our cohort, along with other genes that have previously found amplified in ovarian cancer such as *CCNE1*, *ERBB2*, *RSF1* and deleted genes like *BRCA1*[8,22]. Among the cohort and subset analyses, there were more recurrently deleted

segments than amplified segments and regions within 8q, 3q, and 19q were among the most recurrently amplified segments while peaks on 9q, 15q, 16q, 17q were among the most recurrently deleted segments. We correlated the CNA and our RNA-seq data for all samples in our cohort, utilizing the threshold values GISTIC2.0 calculates with their corresponding RNA-seq FPKM values for the genes involved in altered regions (Supplementary Fig. 2c). The relationship between copy number and expression is not simple, but the medians of the data suggest that more amplified genes trend toward having higher RNA expression.

Our GISTIC2.0 analysis results comparing ST to LT survivors revealed that there were more recurrent copy-altered segments among the ST survivors (ST = 79 amplified, 198 deleted; LT = 60 amplified, 101 deleted). Comparing the primary and metastatic tumor analyses showed that the metastatic tumors had more recurrent segments (primary = 39 amplified, 135 deleted; met= 63 amplified, 157 deleted). Consistent with published data[9], *CCNE1* was amplified in ST survivors, primary tumors, and metastatic tumors sample subsets, but not among the LT survivor samples. Both the primary tumor and metastatic tumor subsets were significantly amplified at 19q12 and at 20q13.12, while the ST subset had 19q12 amplified and the LT subset had 20q13.12 significantly amplified.

**Differentially expressed genes correlate with survival**. We calculated differential expression (DE) of genes between the ST and LT survivors in both primary and metastatic tumors. Overall, there were distinct transcriptomes that correlated with ST or LT survival, both within the tumor cohorts separately and when combining all patients regardless of tumor type (Fig. 3a, b). Within the metastatic tumor cohort, there were 4792 DE genes (DEGs) between ST and LT survivors, with an FDR < 0.01, after selecting for only protein-coding genes. Genes such as *SZRD1* and *ERV3-1* were upregulated in the metastatic tumors of ST survivors, as previously reported in other solid cancer types such as cervical and colorectal cancer[23,24].

In order to identify DEGs that were specifically associated with survival in the metastatic tumors, we filtered out any genes that were also differentially expressed between the ST and LT

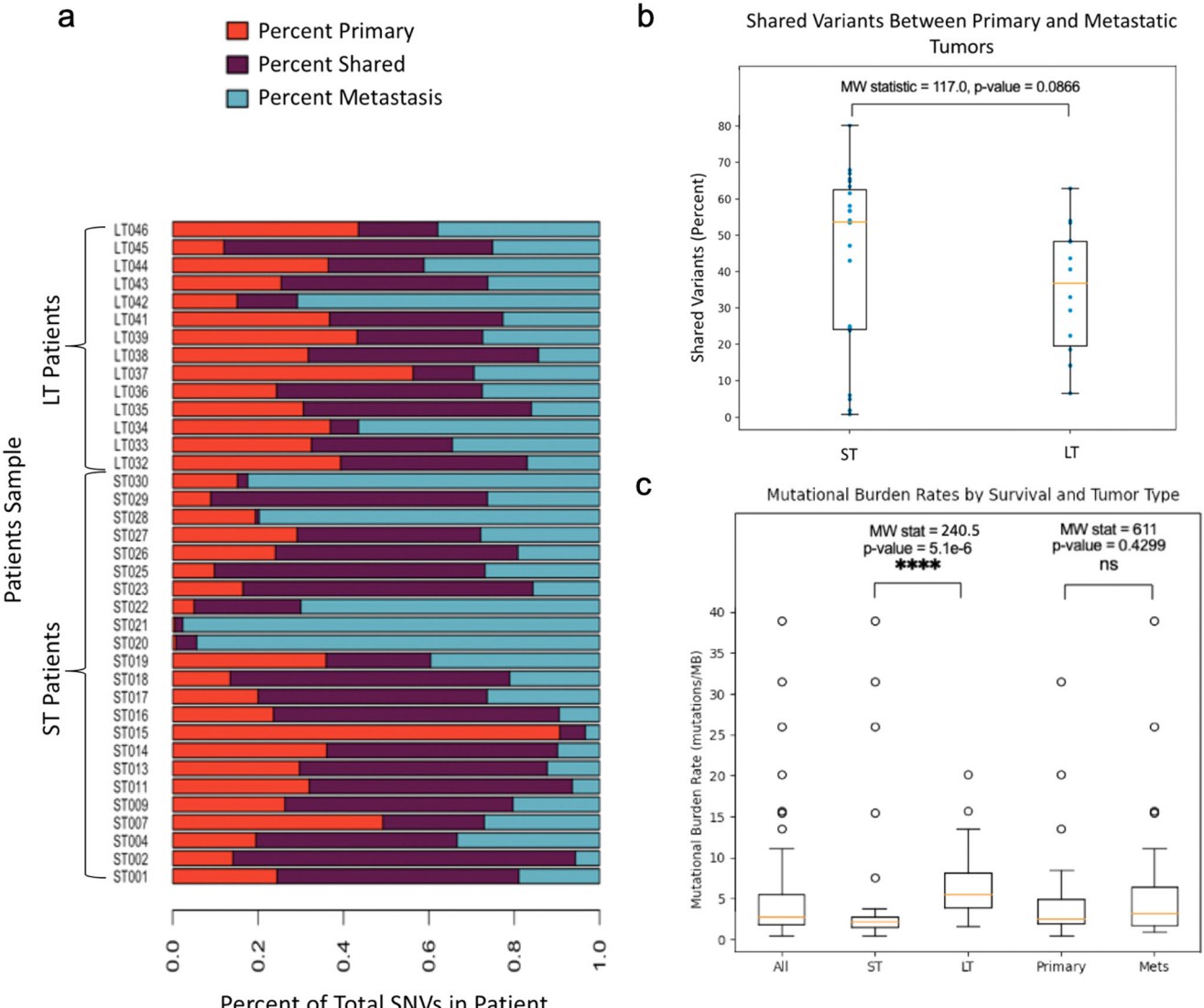

**Fig. 2 Shared variants between primary and metastatic tumors. a** Percentage of somatic mutations unique to primary (Red) and metastatic tumor (Blue) and shared (Purple) between samples. **b** Comparison of percentage of shared variants among primary and metastatic tumor between ST and LT survivors (ST survivors $n = 23$ LT survivors $n = 14$). **c** Boxplot displaying the mutational burden rates (mutations/MB) for all tumors and subsets. (All $n = 74$, ST survivors $n = 23$, LT survivors $n = 17$, primary tumors $n = 37$, metastatic tumors $n = 37$). The boxplots define the range of the minimum to the maximum by lines, a box from the first quartile to the third quartile with the median as the center line. Circles outside the range of the data are outliers.

survivors' primary tumors. This revealed 325 genes only DE in metastatic tumors, with 295 of these (90.7%) downregulated in ST survivors (Fig. 3a). The DEGs unique to metastatic tumors of ST survivors are enriched for several Biological Processes GO terms with an FDR < 0.05, such as "regulation of cellular biosynthetic process" (Supplementary Fig. 6)[25–28]. A GO enrichment analysis on the 30 upregulated DEGs unique to metastatic tumors showed enrichment with an FDR < 0.05 for several Molecular Function GO terms associated with DNA binding and transcriptional activity (Fig. 3c)[25,27,28]. This enrichment is most likely due to the 13/30 of those upregulated DEGs that are in the zinc finger family. We also used DAVID to find enrichment of the KEGG and Biocarta pathways within our DEGs, and although there were no significantly enriched Biocarta terms, there were 9 KEGG pathways enriched. Some of these included "Adherens junction" and "protein processing in endoplasmic reticulum". (Supplementary Fig. 7) Of note, *FOXL2NB* and *PTCH2* have correlated with poor survival in other cancer types[29,30]. There is evidence that

*OGN* plays a role in *EMT*, and *PRDX1* has been studied as prognostic marker in lung cancer[31,32]. We calculated DE between genes of ST and LT survivors within the primary tumors. We found that there was a total of 4248 DE genes with FDR < 0.01. After filtering for protein-coding genes, we narrowed our list of DE genes to 3694, with 502 DEGs that were specifically differentially expressed only in primary tumors (Fig. 3b).

When all tumors are included in the DE analysis, there were a total of 7304 protein-coding DEGs between ST and LT survivors (Supplementary Data 1). The top 50 upregulated and top 50 downregulated DEGs are included in Supplementary Fig. 8. Additionally, we calculated DEGs between primary and metastatic tumors and identified only 4 DEGs with an FDR < 0.01. When we lower the FDR filter to <0.1, the number of DEGs increased to 15. Of those, 5 genes (*WIPF3, STAR, SCUBE1, PEG3, CNTNAP2*) were also found in the top 100 DEGs identified by Sallinen et al., which compared DEGs between 10 matched primary and metastatic ovarian tumors having an FDR < 0.1[12].

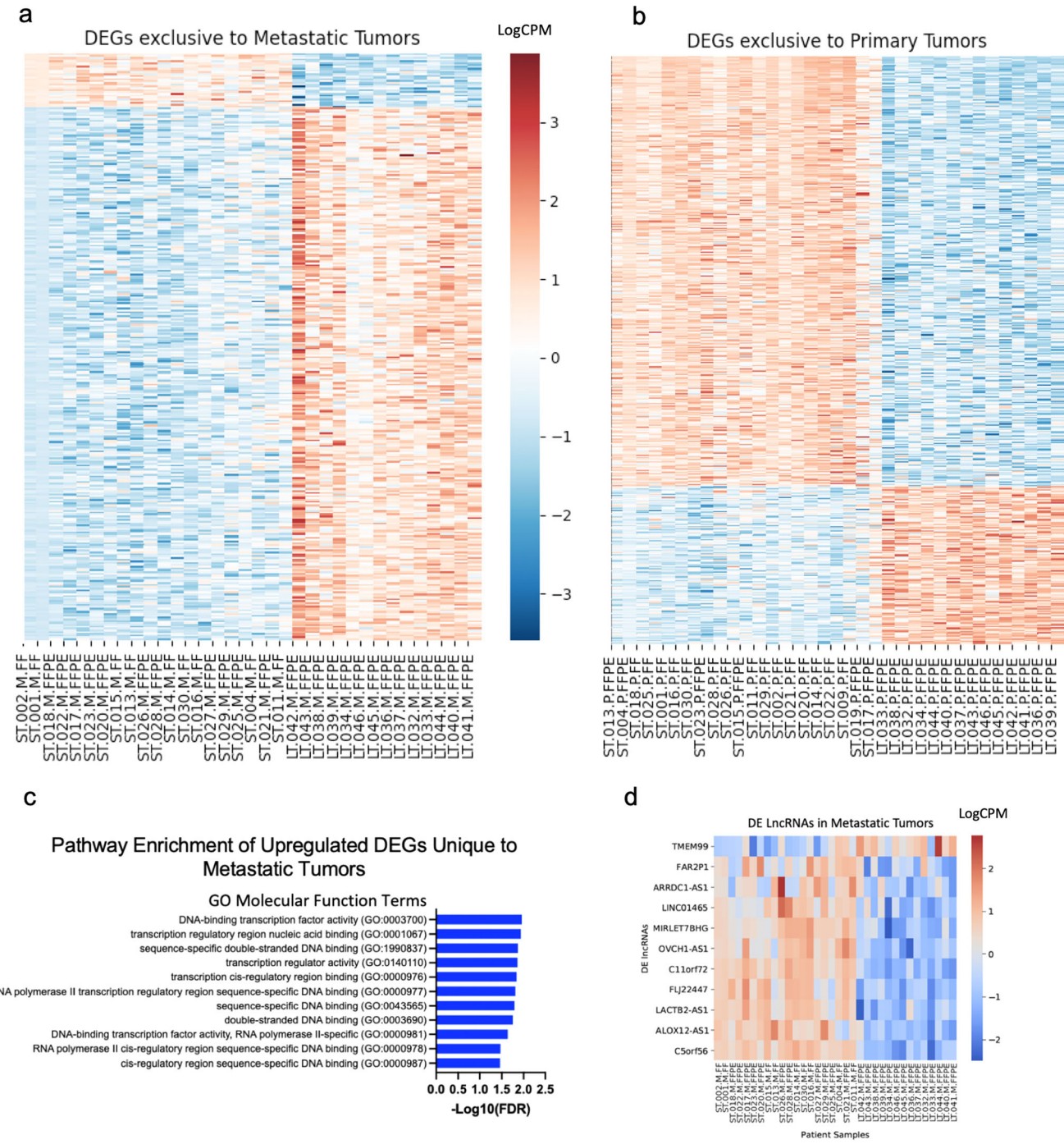

**Fig. 3 DEGs and lncRNAs in tumors from ST and LT survivors. a** Heatmap displaying the 325 significant protein-coding DEGs between ST and LT survivors unique to metastatic tumors in our patient cohort (ST survivors $n = 20$, LT survivors $n = 14$). Red indicates upregulated DEGs while blue indicates downregulated DEGs with FDR < 0.01according to gradient legend. Gene list is provided in Supplementary File 1. **b** Heatmap displaying the 502 significant protein-coding DEGs between ST and LT survivors unique to primary tumors in our patient cohort (ST survivors $n = 21$, LT survivors $n = 14$). Red indicates upregulated DEGs while blue indicates downregulated DEGs with FDR < 0.01according to gradient legend. Gene list is provided in Supplementary File 1. **c** Barplot displaying the GO Molecular Function Terms that are statistically overrepresented in the 30 upregulated DEGs unique to Metastatic tumors. Plot displays their -Log10(FDR) value for each GO term. **d** Heatmap displaying 11 significantly (FDR < 0.01) differentially expressed lncRNAs between the ST and LT survivors in metastatic tumors (ST survivors $n = 20$, LT survivors $n = 14$).

**Differentially expressed lncRNAs correlate with survival.** From the RNA-sequencing data, we identified several long-noncoding RNA transcripts (lncRNAs) that were among the top differentially expressed transcripts between the ST and LT survivors in both metastatic and primary tumors. Within the metastatic tumors, we identified 11 lncRNAs that were differentially

expressed and all but one was upregulated in ST survivors (Fig. 3d). This set of lncRNAs included *ARRDC1-AS1*, which was shown to be a part of a potential lncRNA prognostic signature in breast cancer[33]. Among the primary tumors, we identified 36 lncRNAs of which 35 were upregulated in ST survivors. Of these 36 lncRNAs, 9 lncRNAs (25%), (*FAR2P1, ARRDC1-AS1,*

*MIRLET7BHG, OVCH1-AS1, C11orf72, FLJ22447, LACTB2-AS1, ALOX12-AS1*, and *C5orf56*) overlapped with the lncRNAs identified in our metastatic tumor cohort.

**Tumors from ST survivors harbored recurrent gene fusion predictions.** A total of 1164 gene fusions were predicted among our tumor sample cohort, 35 of which were recurrent (seen in at least 2 samples) and unique to ST survivors (Supplementary Table 6). The higher number of gene fusions identified in ST survivors was due to a higher level of quality in the RNA-sequencing since this subgroup included FF tumor samples, whereas all LT survivors were FFPE samples.

INTEGRATE detected several *ESR1* gene fusions in our tumor cohort, which have previously been implicated in breast and ovarian cancer[9,34]. In particular, the *ESR1 > CCDC170* recurrent gene fusion identified by Yang et al. was present in 2 of our tumor samples, 1 ST primary tumor (5 reads) and 1 LT metastatic tumor (7 reads). Interestingly, we also noticed that a total of 33 gene fusions involved collagen genes, 32 of which were identified in ST survivors, 21 were in metastatic tumors, and 20 are in-frame fusions (Supplementary Table 7). Pathway analysis on the genes involved in recurrent gene fusions in our cohort were significantly enriched for terms related to "collagen chain trimerization" and "Collagen degradation" in the PANTHER reactome set, which is interesting given the known role of collagen in the ovarian cancer tumor microenvironment[25-28].

**Immune cell populations abundances.** We used the program Cibersort to estimate the fraction of immune cell types in our tumor samples (Fig. 4). The immune cell groups CD4 T-Cells, macrophages, and monocytes had the highest fractions in many of the tumor samples. There was much variability in immune cell type fractions across patients, but there were few significant differences in immune cell fractions between primary or metastatic tumors or between ST and LT survivors among the 22 immune cell types (Supplementary Fig. 9). Of note, CD4 naïve T-cells (higher fractions in ST), follicular helper T-cells (higher fractions in LT), regulatory T-cells (higher fractions in LT), and activated dendritic cells (higher fractions in LT) were significantly different between ST and LT survivors with Mann–Whitney statistical p-values < 0.05. Between the primary and metastatic tumors, the CD8 T-cells, activated CD4 memory T-cells, and neutrophils were significantly higher in metastatic tumors based on Wilcoxon statistical p-values < 0.05. A chart of the statistical differences between all of the subsets for the 22 immune cell fractions is in Supplementary Table 8.

Lee et al. found significant abundance differences of M2 macrophages and monocytes between their R0 and NACT patient

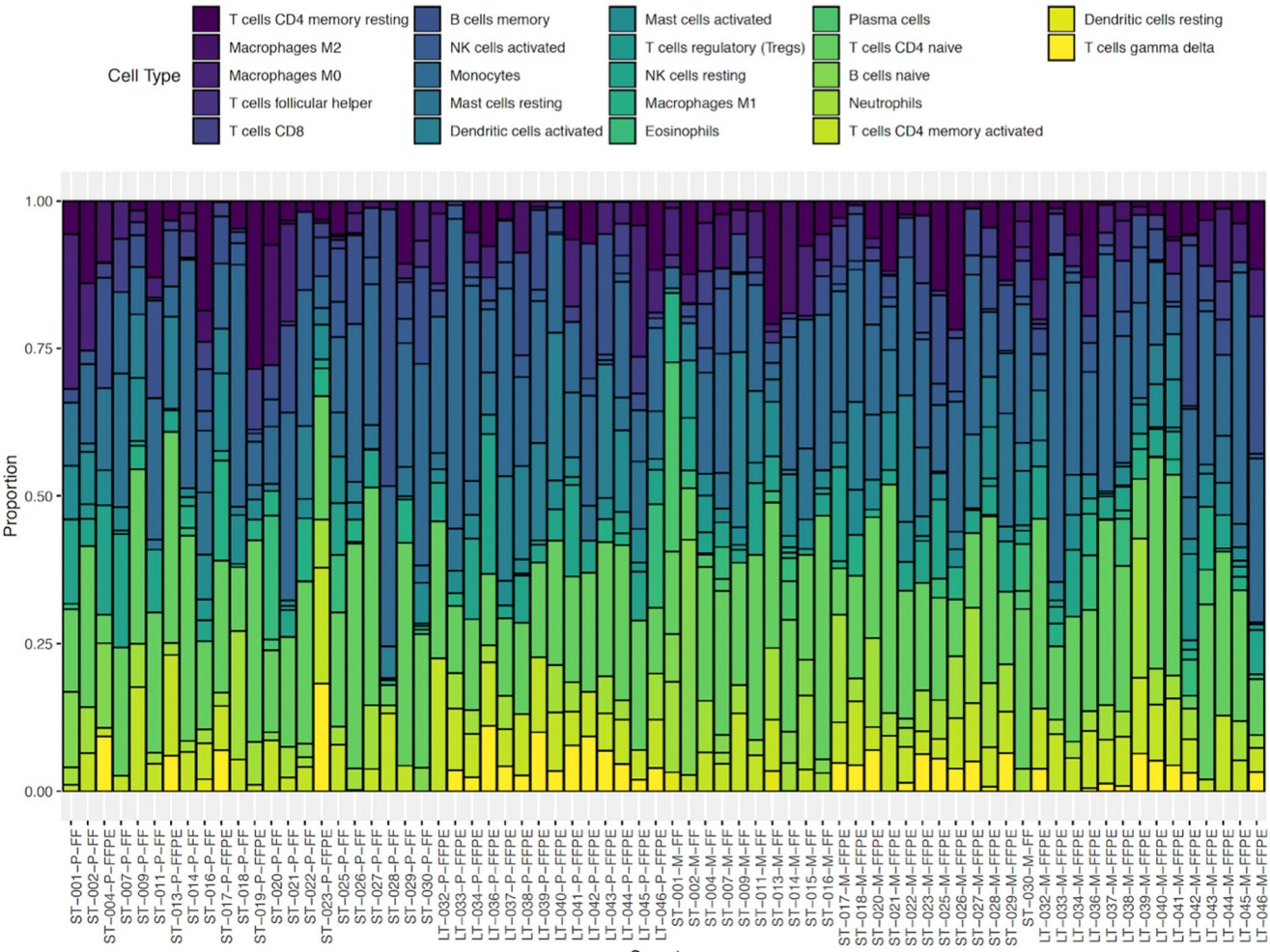

**Fig. 4 Cibersort immune cell fractions stacked barplot of the proportion of the 22 immune cell types expressed in each tumor sample.** Samples are organized by tumor type, then by survival (ST Primary n = 22, LT Primary n = 14, ST Metastatic n = 22, LT Metastatic n = 14). Annotation colors are shown in legend above barplot.

groups, and a significant difference between the abundance of resting CD4 memory T cells between primary and metastatic tumors, but these patterns did not appear in our dataset[11]. Thorsson et al. performed immunogenomic analysis across cancer types in TCGA and identified six immune subtypes[35]. In their analysis, the ovarian cancer cohort correlated the most with their C2 IFN-y dominant signature, which is defined by having high M1 and M2 macrophage polarization and strong CD8 signal. This is consistent with the higher fraction of macrophages we found in our Cibersort analysis. The ovarian cohort also had representation of their C1 wound healing and C4 lymphocyte depleted signatures, but did not have representation for their C3 inflammatory, C5 immunologically quiet, or C6 TGF-B Dominant signatures. The lack of these signatures is consistent with our cohort's low immune cell fractions for several lymphocytes and the variability between patient samples. Since our cohort included metastatic tumors that are not represented in TCGA, perhaps a more specific immunogenomic analysis with more metastatic tumors for ovarian cancer is necessary to better understand the immune landscape in these tumors[35].

## Discussion

HGSC can rapidly metastasize before patients experience symptoms, therefore many patients are diagnosed at late stages and have limited treatment options. Despite many studies of the genetics of HGSC tumors, we have yet to fully characterize and identify genetic biomarkers of HGSC metastatic tumors, especially those with poor survival outcomes. In this study we built on previous studies to better characterize the genomic features of matched primary and metastatic tumors in the context of patient survival, so we might identify unique features of ST metastatic tumors.

We found supporting data for *RB1* mutations as a marker for long survivorship as previously discovered, since *RB1* mutations were identified exclusively in our LT survivor cohort[8,14]. In our study we found that there was a higher percentage of shared variants between the primary and metastatic tumors of ST survivors compared to LT survivors. Although this difference was not significant, it can suggest that tumors from ST survivors may be more clonal and genetically similar than tumors from LT survivors. This could mean that tumors from ST survivors are inherently more resistant to treatments, since both their primary and metastatic tumors are genetically similar. However, other studies of the clonality of HGSC tumors have yet to find a correlative pattern between clonality and survival[1,7,36–38], hence, many more tumor samples will need to be analyzed to answer this question. Shared variants that are likely to be present in all clones of the tumor may be the best suited for targeted therapies. With the advent of single-cell sequencing, we may now be able to answer more questions about the heterogeneity and clonal development of HGSC tumors[39].

*TP53* mutations are a hallmark of high-grade serous ovarian cancer and *TP53* gene has known hotspot mutations across cancer types. One of these hotspot mutations, R273H, was identified in 3 patients within our cohort, two of whom had an overall survival ranging 17-34 months. In TCGA Genomic Data Commons Portal, there are a total of 99 cases across cancer types that harbor a mutation at this position in *TP53*, 9 of which are in ovarian cancer samples. Recent functional studies have shown that this particular mutation results in a p53 gain-of-function that may promote metastasis in colorectal, esophageal, and breast cancers. Additionally, breast cancer cell lines with a R273H gain-of-function have been found to have improved response to combination PARPi and a DNA-damaging agent[40–42]. If this is also seen in ovarian cancer cells, this may lead to additional patients receiving PARP inhibitor and combination treatments in the future. However, further work in characterizing the therapeutic potential of this specific *TP53* mutation in ovarian cancer is needed.

Yang et al. demonstrated genomic differences between HGSC primary tumors of ST and LT extreme survivors[9]. Our study focused on paired primary and metastatic tumors within the median survival range of ovarian cancer. Yang et al. demonstrated that more than 50% of tumors with *BRCA* mutations are LT survivors with an OS > 10 years. This is consistent since HGSC patients with *BRCA* mutations respond better to chemotherapy[10]. Therefore, our study was better able to characterize the genomic features of tumors from patients with more moderate survival to poor survival.

Recently gene fusions have proven to be useful drug targets for cancer. For example, the identification of EML4 > ALK gene fusions in non-small cell lung cancer paved the way for the development of ALK inhibitors and recently drugs targeting tumors of any cancer type with gene fusions involving *NTRK* genes have been approved by the FDA[43,44]. In our analyses, we identified an *ESR1-CCDC170* gene fusion in our cohort as previously described by Yang et al.[9]. There is evidence that *ESR1* gene fusions in estrogen receptor-positive breast cancer promote endocrine therapy resistance and metastasis, thus *ESR1* gene fusions may have a role in ovarian cancer progression[34]. Lei et al. demonstrated that CDK4/6 inhibitors were able to suppress growth that was driven by *ESR1* gene fusions, indicating that gene fusion driven cancers are treatable[34]. We found a higher number of gene fusion predictions in our tumors from ST survivors and these could be a potential source for new drug development, but additional work will be needed to identify recurrent gene fusions that are targetable in ovarian cancer.

Previous studies have demonstrated that HGSC primary and metastatic tumors have similar transcriptomes. Two such studies using microarrays identified few differentially expressed genes between the HGSC primary and metastatic tumors[45,46]. In this study, we also identified few DEGs between primary and metastatic tumors. However, when we analyzed primary or metastatic tumors separately to find DEGs between ST and LT survivors, we found DEGs unique to metastatic tumors from patients with ST survival. This demonstrates that clinical outcome can be used to identify DEGs specific to metastatic tumors. We found several DEGs in the zinc finger family that were upregulated in the ST survivor metastatic, suggesting that these tumors have more transcriptionally active genes that could be promoting metastatsis or could be used as markers for poor prognosis, like FOXL2NB[29] and PTCH2[30], which have correlated with poor survival in other cancer types. The large number of DEGs that identified in our DE analyses are a resource for future studies for biomarkers given their correlation with poor prognosis in ovarian and other cancer types and because we filtered for genes unique to the metastatic tumors in our cohort.

Additionally, we identified lncRNAs that were differentially expressed between survival groups. LncRNAs have only recently been studied for their role in cancer development and prognosis and have not been extensively studied in ovarian cancer yet[47,48]. There are some lncRNAs, such as RP11-190D6.2, that have shown to be tumor suppressors or oncogenes in ovarian cancer cell lines[48,49]. Given that we found several lncRNAs having increased expression in ST survivors, they could serve as potential targets or biomarkers for future treatment development.

It should be noted that all of our LT survivor samples are from FFPE, while ST tumors were not. Though we have applied rigorous quality control and filtering to our variant calling, we cannot exclude the possibility that sample preparation has some effect on the results. It is possible that the batch correction from

FFPE and FF samples reduced the number of DEGs that were able to be identified in our cohort between the primary and metastatic tumors. The SVA batch correction may have over accounted for unknown variation or it may be introduced variation, but was still necessary so we could include all tumor samples in our DE analysis, regardless of RNA sample preparation. This dataset, like many using patient samples, has limitations but provides insights into the differences between HGSC primary and metastatic tumors in the context of moderate survival outcomes.

In conclusion, our research characterizes the exomes and transcriptomes of a unique dataset of matched primary and metastatic tumors in the context of patient survival. We were able to confirm many of the genomic features seen in previous studies[8,9,11,12,16]. We observed that the transcriptomes of primary and metastatic tumors were similar to each other, compared to the transcriptomes of tumors from ST and LT survivors that had more DEGs and DE lncRNAs. Our gene fusion analysis revealed fusions that have the potential to be new targets in HGSC and could warrant further functional studies. In short, our research improves the understanding of genetic variation in HGSC metastases that exist between patients with different prognoses can better inform treatments and may identify new targets for drug development.

## Methods

**Patient cohort sample criteria**. This study was approved by the Washington University in St. Louis Institutional Review Board #201309075). Criteria for approval are met per 45 CFR 46.111 and/or 21 CFR 56.111 as applicable. Patients were included in they had FIGO stage III–IV ovarian cancer of serous ($n = 38$) or endometrioid ($n = 1$) histology and were undergoing primary cytoreductive surgery, and for all patients informed consent was obtained. All research conformed with the principles of the Declaration of Helsinki. We analyzed normal tissue, primary tumor, and metastatic tumor samples from a total of 39 patients.

Normal tissue samples consisted of adjacent non-malignant omentum or peritoneum. All tumors were collected during primary cytoreductive surgery, prior to any chemotherapy treatment, and were stored as either fresh frozen (FF) or formalin-fixed paraffin-embedded (FFPE). These patients were separated into two groups based on their overall survival. Patients who lived less than 3.5 years after their diagnosis were considered short-term (ST) survivors and patients who lived more than 5 years after diagnosis were considered long-term (LT) survivors (Table 1). Other clinical characteristics of patients are shown in Table 1. All patients received standard regimens of carboplatin and paclitaxel following cytoreductive surgery. More LT survivors received intraperitoneal (IP) chemotherapy than ST survivors (1 ST survivors, 5 LT survivors, $p$-value = 0.042), Otherwise there were no differences in the use of bevacizumab or PARP inhibitor treatments between the two cohorts. All 23 ST patients and 12 LT patients had matched DNA and RNA extracted and sequenced. An additional 4 LT patients had tumor sequencing performed: Patients 031 and 035 had matched primary and metastatic tumors DNA sequenced and Patients 032 and 040 only had RNA-sequencing from their matched primary and metastatic tumor tissue.

**Exome and RNA sequencing**. All tumors were examined by a pathologist to determine tumor cellularity and necrosis and only samples of 60% tumor cellularity or higher with <20% necrosis were sequenced. DNA and RNA were extracted from FF or FFPE tissues using Qiagen's DNeasy Blood & Tissue Kit and RNeasy kit. Whole-exome sequencing of DNA from matched primary tumor, metastatic tumors, and normal tissue samples was completed for 39 patients with the NimbleGen VCRome exome capture kit (NimbleGen Roche) according to the manufacturer's protocol. Paired-end Illumina 151 bp reads were generated for normal samples to a minimum of depth of 65x, while tumor samples were sequenced to a minimum of 139x, with the average coverage of ~300x. A coverage table provides per-sample coverage details (Supplementary Table 1). RNA sequencing of primary and metastatic tumor samples was performed using the Illumina TruSeq stranded Total RNA library kit following the Manufacturer-recommended protocol. Paired-end Illumina sequencing of 151 bp read length yielded an average of approximately 125 million paired reads per-sample and an average of approximately 134 million reads mapped per sample. Quality Control metrics for the RNA-seq samples were generated using MultiQC and are reported in Supplementary Data 2 [50].

**Variant calling and genomic analysis**. Exome sequencing data were aligned to human reference build GRCh37 using BWA-mem and deduplicated with Picard version 1.113[51,52]. Somatic variants were called from combined data using the Genome Modeling System pipeline[53,54]. In brief, variants were called from the

## Table 1 Clinical characteristics of patient cohort.

|  | Short-term (ST) | Long-term (LT) |
|---|---|---|
| Patients (no.) | $n = 23$ | $n = 16$ |
| Tumors | 46 | 32 |
|   Primary | 23 | 16 |
|   Metastatic | 23 | 16 |
| Age (years) | 61.5 ± 19.5 | 57 ± 8.2 |
| FIGO stage |  |  |
|   IIIA | 2 | 0 |
|   IIIC | 14 | 16 |
|   IV | 7 | 0 |
| FIGO grade |  |  |
|   Moderately differentiated | 3 | 0 |
|   Poorly differentiated | 20 | 16 |
| Histology |  |  |
|   Serous | 22 | 16 |
|   Endometrioid | 1 | 0 |
| Median overall survival (OS) (months) | 21 (range: 0–41) | 111 (range: 82–195) |
| Fresh frozen (FF) samples |  |  |
|   Primary | 17 | 0 |
|   Metastatic | 11 | 0 |
| Paraffin-fixed (FFPE) samples |  |  |
|   Primary | 6 | 16 |
|   Metastatic | 12 | 16 |
|   Whole-exome sequenced | 46 | 14 |
|   RNA-sequenced | 46 | 14 |

Clinical characteristics of patients diagnosed with HGSC at FIGO stage III-IV. Tumor samples were collected from patients during primary cytoreductive surgery in Washington University in St. Louis.

union of 4 callers, which included Samtools version r932, Somatic Sniper version 1.0.4, VarScan version 2.3.6, Strelka version 1.0.11, and Mutect v1.1.4[53,55–59]. Indels were detected from the union of 4 callers; GATK somatic-indel version 5336, Pindel version 0.5, VarScan version 2.3.6, and Strelka version 1.0.11[53,55,56,60,61]. Further variant filtering was applied as described in Ghobadi et al.[53]. Briefly, SNVs and indels were discarded if they had below 20x coverage, appeared as artifacts in a panel of 905 normal exomes, or exceeded 0.1% frequency in the 1000 genomes or NHLBI exome sequencing projects[62,63]. A Bayesian classifier (https://github.com/genome/genome/blob/master/lib/perl/Genome/Model/Tools/Validation/IdentifyOutliers.pm) was also applied and variants that classified as somatic with a binomial log-likelihood of at least 5 were retained[53]. All called variants compared in this study are provided in Supplementary Data 2. Mutational burden was calculated as the number of variants called per megabase for all variants that passed the QC filtering. The waterfall plot (Fig. 1a, b) depicting frequently mutated genes from TCGA-OV was generated using GenVisR[8,64]. Mutational clinical significance for somatic and germline BRCA mutations was determined from ClinVar using their definitiions of clinical significance terms (https://www.ncbi.nlm.nih.gov/clinvar/) (Supplementary Data 2, Supplementary Tables 3 and 4)[15]. Classification of Ovarian Cancer signatures were calculated according to parameters defined in Verhaak et al.[16]. Copy-altered segments were identified from VarScan (Supplementary Figs. 1 and 2)[56]. Significant copy-altered segments were identified for all tumors, all tumors from ST survivors, all tumors from LT survivors, metastatic tumors, primary tumors, and only the metastatic tumors of ST survivors using the GISTIC 2.0 version 6.15.28 Module on the AWS GenePattern cloud (https://cloud.genepattern.org/gp). Default parameters and reference genome Human_Hg19.mat were used to run GISTIC 2.0 analyses[21]. We used the wide peak region analyses from GISTIC 2.0 to calculate the total number of genes amplified or deleted within those regions. The correlation between copy number alteration (CNA) and RNA-seq expression was completed using the thresholded CNA values GISTIC 2.0 calculated based on each sample's segment files[21,65]. The violin plot was created by binning all CNA threshold values from every gene for every sample and plotting that with their corresponding log2(FPKM) values (Supplementary Fig. 2c).

### Differential expression analysis

*Normalization and quality control*. Transcript read counts were obtained using Kallisto version: v0.43.1 and gene-level read counts were calculated using GRCh37 in Ensembl[66]. Quality control and normalization of the raw count data were performed using the R/Bioconductor package edgeR version 3.28[67]. For our comparison of LT survivor samples to ST survivor samples, we removed genes with less than 1 Count Per Million mapped reads in at least half of the samples to ensure that a gene was retained if expressed in only one of the two groups. For our comparison of primary to metastatic tumors, genes with less than 1 Count Per

Million mapped reads in at least half the samples were removed. Normalization factors were calculated using the Trimmed Mean of *M*-values normalization method in edgeR to account for compositional biases in libraries between each pair of samples.

*Removal of batch effects.* Due to technical artifacts introduced by the use of FFPE that can affect gene expression analyses, we performed batch effect correction prior to differential expression analysis for the comparison of LT to ST survival samples[68,69]. We used the SVA function of the R/Bioconductor package SVA version 3.34.0 to estimate and remove surrogate variables for unwanted and unknown batch effects and other sources of variation present in the data[70]. The SVA function estimated surrogate variables for each subset analysis, which was adjusted for within the statistical model applied in the edgeR package in downstream analyses of differential gene expression. After batch effect correction, samples were analyzed by a Principal Component Analysis using the R function "dist" on regularized log-transformed (rlog) data to calculate the Euclidian distance between samples. Plotting of the first (PC1) and second (PC2) principal components revealed that expression values from the same patient are more related to one another than between groups (Supplementary Fig. 3*)*. We also observed 4 potential outlier samples, which were removed from downstream analyses because of their distance from the other samples in the Principal Component Analysis plot after normalizing and batch correcting transcript counts. These 4 removed samples are highlighted in Supplementary Fig. 3a and were all collected within the same year, but their exclusion could mean we are missing out on some biological features of these tumor samples.

*Differential gene expression (DGE) analysis.* DGE analysis was performed using edgeR version 3.28.0, which implements a negative-binomial general linear model[67]. We performed 4 comparisons: ST survival samples versus LT survival samples for all tumors in the study; ST survival versus LT survival among metastatic tumors; ST survival versus LT survival among primary tumors; and primary tumors versus their matched metastatic tumor. The surrogate variables estimated with SVA were included in the model used for the LT versus ST survival comparison. To normalize gene-level variance, the biological coefficient of variation was calculated using Cox-Reid dispersion for negative-binomial general linear models. The *p*-values of differential expression tests were corrected for multiple-hypothesis testing using Benjamini–Hochberg false-discovery rate (FDR) correction. The threshold for significance was set to FDR *Q*-value < 0.01. We further curated our differentially expressed genes (DEGs) by limiting to protein-coding genes that were listed in Ensembl genes 100 Human genes (GRCh38.p13) protein_coding transcript type on BioMart. All DEGs discussed in this paper are listed in Supplementary Data 1. Pathway analysis was applied to the DEG and gene fusion gene lists using the PANTHER classification system 16.0 (http://pantherdb.org/), with the organisms set as 'Homo sapiens' and performing a statistical overrepresentation test using Fisher's Exact test and calculating a False-Discovery Rate[25]. We used all Gene Ontology (GO) terms (Biological Processes, Molecular Function, and Cellular Components), PANTHER pathways, and Reactome pathways annotation sets[26–28]. We used DAVID to identify enrichment for KEGG and Biocarta pathways[71,72].

**Immune cell abundance estimates**. We used Cibersort (https://cibersort.stanford.edu/) to estimate the abundance of infiltrating immune cell types using our tumor RNA-seq data[73]. We generated a mixture file for our cohort of tumor samples based on the gene abundance counts generated from the RNA-seq reads using Kallisto[66]. We used the LM22 gene signature, which calculated immune cell fractions for immune cell types, and ran our Cibersort analysis with 500 permutations under the relative mode.

**Gene fusion predictions**. Gene fusion predictions for each tumor sample were produced using INTEGRATE v0.2.6 to analyze the tumor RNA-sequencing data[74]. Full-length raw reads and a set of reads trimmed to remove potentially low-quality bases were each aligned to human reference genome GRCh38 (r90) using STAR v2.5.3a with a minimum chimeric segment length of 18 and chimeric alignments output to a separate SAM file[75]. The chimeric alignments were then used as inputs for INTEGRATE fusion with default parameters for fusion discovery with tumor RNA-seq only. Fusion predictions from the full and trimmed reads were then merged and manually reviewed to ensure all fusion calls were valid. Since normalization of FFPE and FF tumor samples is more challenging for gene fusions, we characterized the predicted gene fusions as independent events regardless of sample preparation.

**Statistics and reproducibility**. We analyzed normal tissue, primary tumor, and metastatic tumor samples from a total of 39 patients. Statistical analysis and figure generation was performed in R 3.6.2 and Python 3.8.2. The *p*-values of differential expression tests were limited to an FDR *Q*-value < 0.01. Comparisons between survival groups were determined by Mann–Whitney *U*-statistical tests for individual samples and comparisons between tumor types were performed using Mann–Whitney–Wilcoxon statistical tests for dependent samples since the primary and metastatic tumors were matched. Enrichment for pathway analyses with our DEGs was done using Fisher's Exact test and calculating a false-discovery rate.

Given the genetic heterogeneity of individuals and their tumors, it should be noted that we sequenced only one sample from each primary and metastatic tumor, which limits our abilities to fully capture the genetic diversity within these tumors.

**Reporting summary**. Further information on research design is available in the Nature Portfolio Reporting Summary linked to this article.

## Data availability

RNA-sequencing files have been deposited in the NCBI GEO data base under GSE218939. WES data generated for this analysis have been deposited within the Sequence Read Archive under the accession PRJNA957243, and can be found at https://www.ncbi.nlm.nih.gov/bioproject/PRJNA957243. Lists of SNVs, DEGs, lncRNAs, and gene fusions are provided in Supplementary Data 1 and 2. Source data for figures have been submitted in Supplementary Data 3.

## Code availability

Code used to analyze genomic data is publicly available and custom code is deposited on Github (https://github.com/ekotnik/OC-Tumor-genomic-analyses) https://doi.org/10.5281/zenodo.7873762[76].

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

## Acknowledgements

The authors are grateful to all the of the participants in this study as well as the surgeons and supporting staff that made sample collection and sequencing possible. We'd like to acknowledge Dr. John Edwards for providing insight, guidance, and comments for the duration of this project and manuscript preparation. We'd also like to thank Megan Richters for her insight, support, and advice with genomic analyses. Funding for this study was provided by the Foundation for Barnes-Jewish Hospital.

## Author contributions

E.N.K. contributed to consolidating, analyzing, interpreting all genomics data, creating figures and writing and preparing the manuscript. M.M.M. contributed to interpretation of the data and collecting patient clinical data. N.C.S. contributed to analyzing and interpreting the SNV and copy number data. T.L. contributed to initial tumor sequencing processing and genomic data analysis. M.I. and J.Z. analyzed RNA-seq data with INTEGRATE for gene fusion analysis. F.M-R. contributed code and helped with DE analysis. I.S.H., C.K.M., P.H.T., A.R.H., M.A.P., and D.G.M. all contributed to surgically collecting tumors and patient data collection for the study. C.A.M. helped interpret gene fusion and genomic data. C.A.M., E.R.M., and G.L. contributed to advising on genomic analyses, interpreting genomic data, and revising this manuscript. D.K. contributed to revising this manuscript. K.C.F. is the senior author and contributed to the conception, design of the project, and preparation of the manuscript. The authors have read and approved of the final manuscript.

## Competing interests

The authors declare no competing interests.

## Consent for publication

All patient information was anonymized and no identifying personal information was collected.
