## [Peer Review File · Communications Biology]

Genetic Characterization of Primary and Metastatic High-Grade Serous Ovarian Cancer Tumors Reveals Distinct Features with SurvivalReviewers' comments:

Reviewer #1 (Remarks to the Author):

The authors have substantially improved their manuscript and answered all the concerns. I have no more comments and I congratulate the authors for their work.

Reviewer #3 (Remarks to the Author):

Many thanks to the authors for their extensive alterations in response to my review, I have no further concerns. The authors have been transparent about the potential limitations of their study and have included relevant supplementary data for readers to interrogate.

Reviewers #4-5 (Remarks to the Author):

In this manuscript, Kotnik et al., analyzed genomic and transcriptomic alterations in paired primary and metastatic tumors for short-term (ST) and long-term (LT) high-grade ovarian cancer survivors. They compared somatic variants including copy number alterations, differential gene expression, fusions, and immune cell infiltration patterns between the groups. The study found greater differences between ST vs. LT cancers than between paired primary and metastatic tumors.

Several pertinent points were raised by Reviewer #2, and we have listed our feedback based on the author's point-by-point responses below.

1. The revised manuscript was edited to clarify actual number of samples collected and sequenced. However, there still several discrepancies. The authors response state 39 patients were sequenced (lines 121-122). Lines 112-115 state that 35 (23 ST + 12 LT) patients were sequenced. Table 1 states that all 46 ST samples from 23 patients were sequenced but only 14 out of possible 32 samples from 16 patients were sequenced. Fig. 1 and 2 state that 23 ST + 14 LT samples were sequenced.

The numbers described in different sections of the manuscript still do not corroborate and the authors must very carefully describe the actual numbers of samples analyzed. In its current state, these errors lead to concerns about the reliability of the reported results.

Moreover, several RNAseq samples were excluded from downstream analyses based on their appearance as outliers on the PCA-plots. Unless there is a strong justification for exclusion based on technical reasons (for example, low depth of coverage, high rRNA content or other evidence of sequencing or technical bias), it is unusual to drop biological independent samples from the analyses. This may lead to loss of important biologically relevant signals or bias the results.

2. The description of the mutation analyses was adequately revised to state the type of signatures used in the analyses.

3. Figure 1 was edited to clearly show the distinction between ST and LT samples. However, this figure provides no information beyond prevalence of TP53 mutations in its current form. Perhaps the authors can extend their analyses to determine somatic variants that differ between ST and LT tumors or between primary and metastatic. Further, an enrichment test for the mutation signatures can be performed to demonstrate how they are different between the groups. We are assuming that primary and metastatic samples are displayed in pairs, but this should be clearly stated in the legend.

The panels 2B and 2C were updated to include results from U-tests – the authors should check the accuracy of their statistical tests as U statistic for both comparisons in 2C is the same but with

different p-values.

4. The authors clarified that the amplification peaks were observed on 19q and loss peaks were on 9q.
5. The discussion section was updated to mention the importance of lncRNAs.
6. Figures and tables were updated to include GO enrichment terms.

While the authors addressed specific comments 5 and 6 for the two RNAseq based analyses, we find that the analyses were inadequate and offer little insight into the biological differences between the groups. First, the analyses should be performed using survival groups and sample sources as covariates, as these are not independent samples. Second, the pathway enrichment analyses should cover a broader range of molecular signatures (for instance, using MSigDB signatures for hallmark and other curated signatures) to provide a better understanding of the differences between the groups.

7. The authors have updated the results and supplementary information to provide requested information on the Cibersort analyses.

The manuscript collected data from several samples that were sequenced using RNAseq and WES. However, we agree with the reviewer that the reported analyses provide little new information or critical insights beyond what is already known, in general, about the disease.

The de-identified data from these analyses must be placed in a database repository if this paper is to be published, such as dbGAP. The authors state that raw data release was not covered by patient consent, while also stating that all data analyzed is "freely available" upon request. Posting just the DE centered values for RNA-seq data is not acceptable (File 1). Further, File 2 is relatively illegible. As the data are presented, the results are not reproducible. As the processed and de-identified data, including all somatic tumor variants, amplification/deletions, indels, etc, and RNA-seq summarized data, are not identifiable, that data should be posted. Further, the custom code should be deposited in the Github repository prior to review.

Genetic Characterization of Primary and Metastatic Tumors with Survival in High-Grade Serous Ovarian Cancer

Nature Communications Biology Resubmission

Response to Reviewers

Reviewer #1 (Remarks to the Author):

The authors have substantially improved their manuscript and answered all the concerns. I have no more comments and I congratulate the authors for their work.

Reviewer #3 (Remarks to the Author):

Many thanks to the authors for their extensive alterations in response to my review, I have no further concerns. The authors have been transparent about the potential limitations of their study and have included relevant supplementary data for readers to interrogate.

Reviewers #4-5 (Remarks to the Author):

In this manuscript, Kotnik et al., analyzed genomic and transcriptomic alterations in paired primary and metastatic tumors for short-term (ST) and long-term (LT) high-grade ovarian cancer survivors. They compared somatic variants including copy number alterations, differential gene expression, fusions, and immune cell infiltration patterns between the groups. The study found greater differences between ST vs. LT cancers than between paired primary and metastatic tumors.

Several pertinent points were raised by Reviewer #2, and we have listed our feedback based on the author's point-by-point responses below.

Comment 1A: The revised manuscript was edited to clarify actual number of samples collected and sequenced. However, there still several discrepancies. The authors response state 39 patients were sequenced (lines 121-122). Lines 112-115 state that 35 (23 ST + 12 LT) patients were sequenced. Table 1 states that all 46 ST samples from 23 patients were sequenced but only 14 out of possible 32 samples from 16 patients were sequenced. Fig. 1 and 2 state that 23 ST + 14 LT samples were sequenced.

The numbers described in different sections of the manuscript still do not corroborate and the authors must very carefully describe the actual numbers of samples analyzed. In its current state, these errors lead to concerns about the reliability of the reported results.

Response 1A: We appreciate this comment and have edited this to provide more clarity. Sequencing was performed on 39 patients in which 23 were ST patients and 16 were LT patients. We currently state in lines 112-115 that there are a total of 23 ST patients with matched primary and metastatic tumors and 12 LT patients with matched primary and metastatic tumors both with RNA and WES. The additional 4 LT patients have a combination of WES and RNA sequencing performed, but not both methods for their matched tumors. So there are a total of 14 LT patients with matched samples with RNA sequencing, and a total of 14 LT patients with

matched tumors that had WES, but these are not the exact same patient IDs for the LT tumor WES and RNA sequencing cohorts. This has been edited in lines 112-115.

Comment 1B: Moreover, several RNAseq samples were excluded from downstream analyses based on their appearance as outliers on the PCA-plots. Unless there is a strong justification for exclusion based on technical reasons (for example, low depth of coverage, high rRNA content or other evidence of sequencing or technical bias), it is unusual to drop biological independent samples from the analyses. This may lead to loss of important biologically relevant signals or bias the results.

Response 1B: Thank you for this. We have now included the additional PCA plot as Supplementary Figure 3 that displays the 4 outlier samples. The findings from the PCA plot led to exclusion of these from the DE analysis. There was significant distance of these outliers, and this is the basis for not including these in the DE analysis. The samples removed were the primary and met from patient 007, the primary tumor from patient 027 and the met tumor from patient 009. All of these tumor samples were the only ones collected and prepared in 2001 and are ST survivor tumor samples. We elevated for read depth or QC errors with RNA sequencing files, but we did not identify any technical errors (see RNA-seq QC parameters in Supplementary Table 2). After incorporating the SVA batch correction in our DE analysis, there was only a difference in 33 genes, therefore we do not believe that including the batch correction led to loss of biologically relevant signals or bias. We did include in lines 193-195 that there is still a possibility that we are losing some biological signals from this exclusion.

Comment 2: The description of the mutation analyses was adequately revised to state the type of signatures used in the analyses.

Comment 3: Figure 1 was edited to clearly show the distinction between ST and LT samples. However, this figure provides no information beyond prevalence of TP53 mutations in its current form. Perhaps the authors can extend their analyses to determine somatic variants that differ between ST and LT tumors or between primary and metastatic. Further, an enrichment test for the mutation signatures can be performed to demonstrate how they are different between the groups. We are assuming that primary and metastatic samples are displayed in pairs, but this should be clearly stated in the legend.

Response 3: When first analyzing this data, we did analyze differences in the SNVs between the primary and metastatic tumors and between the ST and LT survivors. These analyses did not yield any significant results that would add to the literature. We did not observe any patterns or differences that we were confident were not due to a small cohort or passenger mutations. Ovarian cancer is not a mutationally driven cancer and mostly has greater changes in its copy number alterations, likely due to *TP53* mutations that are prevalent in virtually all high-grade serous ovarian cancer tumors. Instead, we chose to highlight the recurrently mutated genes identified from the TCGA analysis and described the mutational differences between primary and metastatic tumors within patients of our cohorts, as described in figure 2.

In terms of an enrichment test, we have described in lines 265-273 which mutational signatures were the most prevalent in our cohorts and show the comparison of mutational signatures

percentages between ST and LT and between primary and metastatic tumors with box and whisker plots in Supplementary Figure 1 panels B and C.

We have now added this to the legend of Figure 1 so that the samples are organized as tumor pairs on lines 698-699.

Comment 3B: The panels 2B and 2C were updated to include results from U-tests – the authors should check the accuracy of their statistical tests as U statistic for both comparisons in 2C is the same but with different p-values.

Response 3B: We appreciate the statistical test comment, and we have checked the accuracy of the statistical tests. The Mann Whitney-Wilcoxon Test stat between the ST and LT survivors should have been stated as 240.5, and the p-value is correct. This has been edited in the graph in Figure 2 panel C.

Comment 4: The authors clarified that the amplification peaks were observed on 19q and loss peaks were on 9q.

Comment 5: The discussion section was updated to mention the importance of lncRNAs.

Comment 6: Figures and tables were updated to include GO enrichment terms.

While the authors addressed specific comments 5 and 6 for the two RNAseq based analyses, we find that the analyses were inadequate and offer little insight into the biological differences between the groups. First, the analyses should be performed using survival groups and sample sources as covariates, as these are not independent samples. Second, the pathway enrichment analyses should cover a broader range of molecular signatures (for instance, using MSigDB signatures for hallmark and other curated signatures) to provide a better understanding of the differences between the groups.

Response 6: We appreciate this comment and have focused this analysis on the DEGs between primary versus metastatic tumors in ST vs LT survivors. This allows us to identify the unique features for metastatic tumors and primary tumors. In terms of pathway enrichment analyses, we performed enrichment tests with GO terms using Panther for those DEGs. To provide a more thorough enrichment analysis, we used DAVID to perform enrichment with more gene sets, beyond the GO term gene sets provided by Panther. We performed multiple combinations of DEGs with all of the PANTHER GO term gene sets. However, the terms or gene sets that were identified did not have significant FDR values. With DAVID, we focused on the KEGG and Biocarta pathway gene sets and identified some significant pathways for KEGG (see below). However, we did not identify any pathways from the Biocarta set. The KEGG pathways, as seen below, have been added to Supplementary Figure 4 as panel C and edited into lines 215-216, 363-366, and 804.

Sublist	Category	Term	RT	Genes	Count	%	P-Value	Fold Enrichment	Benjamini	FDR	Fisher Exact
[ ]	KEGG_PATHWAY	Adherens junction	RT		6	1.9	5.2E-3	5.3	6.2E-1	6.2E-1	8.9E-4
[ ]	KEGG_PATHWAY	Protein processing in endoplasmic reticulum	RT		9	2.8	5.6E-3	3.3	6.2E-1	6.2E-1	1.6E-3
[ ]	KEGG_PATHWAY	SNARE interactions in vesicular transport	RT		4	1.3	1.5E-2	7.6	1.0E0	1.0E0	1.8E-3
[ ]	KEGG_PATHWAY	Alcoholic liver disease	RT		7	2.2	2.5E-2	3.1	1.0E0	1.0E0	7.4E-3
[ ]	KEGG_PATHWAY	mTOR signaling pathway	RT		7	2.2	3.7E-2	2.8	1.0E0	1.0E0	1.2E-2
[ ]	KEGG_PATHWAY	mRNA surveillance pathway	RT		5	1.6	6.7E-2	3.2	1.0E0	1.0E0	1.9E-2
[ ]	KEGG_PATHWAY	Phospholipase D signaling pathway	RT		6	1.9	8.5E-2	2.5	1.0E0	1.0E0	3.1E-2
[ ]	KEGG_PATHWAY	Insulin resistance	RT		5	1.6	9.1E-2	2.9	1.0E0	1.0E0	2.9E-2
[ ]	KEGG_PATHWAY	MAPK signaling pathway	RT		9	2.8	9.4E-2	1.9	1.0E0	1.0E0	4.5E-2

Comment 7A: The authors have updated the results and supplementary information to provide requested information on the Cibersort analyses.

The manuscript collected data from several samples that were sequenced using RNAseq and WES.

However, we agree with the reviewer that the reported analyses provide little new information or critical insights beyond what is already known, in general, about the disease.

Response 7A: We appreciate this comment and will keep this Cibersort analyses to provide confirmatory information.

Comment 7B: The de-identified data from these analyses must be placed in a database repository if this paper is to be published, such as dbGAP. The authors state that raw data release was not covered by patient consent, while also stating that all data analyzed is “freely available” upon request. Posting just the DE centered values for RNA-seq data is not acceptable (File 1). Further, File 2 is relatively illegible. As the data are presented, the results are not reproducible. As the processed and de-identified data, including all somatic tumor variants, amplification/deletions, indels, etc, and RNA-seq summarized data, are not identifiable, that data should be posted. Further, the custom code should be deposited in the Github repository prior to review.

Response 7B: The RNA-seq files have now been deposited in NCBI GEO under GSE218939 and will be made publicly-available upon publication (it is currently set for December 1, 2023 but this release date can be changed). The raw RNA-sequencing data files are submitted to SRA through GEO. We are currently submitting the WES files to dpGaP and will be deposited there upon their approval of this non-NIH funded data. All variants/indels and copy number alterations are already available in the supplemental figures and files. Custom code for the RNA-sequencing analysis have been deposited on Github and can be found at <https://github.com/ekotnik/OC-Tumor-genomic-analyses>. All germline data, as it is identifiable, cannot be deposited in public databases. The lines describing where this data has been deposited are now detailed on lines 593-600. If reviewers or editors have further concern about these analyses and their reproducibility, we are happy to comply with any other data/code they deem necessary for the publication of this work.

Our previous reviewer 2 commented that all SNVs should be provided in the supplement, and as we have supplied our file used for these analyses that lists all SNVs identified, as described in the methods. We are happy to comply with any format changes reviewers or editors suggest so the data is accessible as possible.

REVIEWERS' COMMENTS:

Reviewer #5 (Remarks to the Author):

The edits to the manuscript in response to my comments are satisfactory and I commend their efforts to make the data publicly available. The authors have addressed all my concerns and the manuscript is suitable for acceptance.